# The Bile Acid Membrane Receptor TGR5 in Cancer: Friend or Foe?

**DOI:** 10.3390/molecules27165292

**Published:** 2022-08-19

**Authors:** Youchao Qi, Guozhen Duan, Dengbang Wei, Chengzhou Zhao, Yonggui Ma

**Affiliations:** 1Department of Veterinary Medicine, College of Agriculture and Animal Husbandry, Qinghai University, Xining 810016, China; 2Academy of Agriculture and Forestry Sciences, Qinghai University, Xining 810016, China; 3Tibetan Medicine Research Center, Tibetan Medicine College, Qinghai University, Xining 810016, China; 4State Key Laboratory of Plateau Ecology and Agriculture, Qinghai University, Xining 810016, China; 5Key Laboratory of Medicinal Animal and Plant Resources of Qinghai Tibetan Plateau, Qinghai Normal University, Xining 810008, China; 6Academy of Plateau Science and Sustainability, Qinghai Normal University, Xining 810008, China

**Keywords:** TGR5, bile acids, cancer, oncogenic functions

## Abstract

The G-protein-coupled bile acid receptor, Gpbar1 or TGR5, is characterized as a membrane receptor specifically activated by bile acids. A series of evidence shows that TGR5 induces protein kinase B (AKT), nuclear factor kappa-B (NF-κB), extracellular regulated protein kinases (ERK1/2), signal transducer and activator of transcription 3 (STAT3), cyclic adenosine monophosphate (cAMP), Ras homolog family member A (RhoA), exchange protein activated by cAMP (Epac), and transient receptor potential ankyrin subtype 1 protein (TRPA1) signaling pathways, thereby regulating proliferation, inflammation, adhesion, migration, insulin release, muscle relaxation, and cancer development. TGR5 is widely distributed in the brain, lung, heart, liver, spleen, pancreas, kidney, stomach, jejunum, ileum, colon, brown adipose tissue (BAT), white adipose tissue (WAT), and skeletal muscle. Several recent studies have demonstrated that TGR5 exerts inconsistent effects in different cancer cells upon activating via TGR5 agonists, such as INT-777, ursodeoxycholic acid (UDCA), and taurolithocholic acid (TLCA). In this review, we discuss both the ‘friend’ and ‘foe’ features of TGR5 by summarizing its tumor-suppressing and oncogenic functions and mechanisms.

## 1. Introduction

G-protein-coupled receptors (GPCRs) are a large family of receptors that are abundantly distributed in different tissues in mammals, with numerous significant effects, such as anti-inflammatory effects, immune regulation, regulation of energy metabolism, regulation of glucose metabolism, anti-aging effects, protection against radiation, anti-cancer effects, and neuroprotection [1,2,3,4]. GPCRs have the same seven transmembrane domains on the cell membrane [5]. Upon GPCRs being activated by ligands of drugs, lipids, and hormones, extracellular signals are transduced into intracellular signals by cascade amplification, thereby exerting important effects and regulating physiological and pathological processes in human beings and animals [6]. These key roles of GPCRs are related to cellular signaling pathways, such as protein kinase B (AKT), nuclear factor kappa-B (NF-κB), extracellular regulated protein kinases (ERK1/2), signal transducer and activator of transcription 3 (STAT3), and cyclic adenosine monophosphate (cAMP) [7,8].

The G-protein-coupled bile acid receptor (Gpbar1), also called the TGR5 receptor or M-BAR, belongs to the family of GPCRs. It is reported that TGR5 is widely expressed in different cells and tissues, for instance in the endocrine gland, adipocytes, muscles, immune organs, spinal cord, and the enteric nervous system [9]. In recent years, a large number of findings have revealed that TGR5 is not only the second bile acid receptor, but also a metabolic regulator taking part in energy homeostasis, bile acid homeostasis, and glucose metabolism via NF-κB, AKT, and ERK signaling pathways [10,11,12]. Upon the binding of ligands in TGR5 domains, activated TGR5 transduces the special extracellular signal into an intracellular signal to regulate downstream cascades, such as 6α-ethyl-23(*S*)-methyl-cholic acid (6-EMCA, INT-777), chenodeoxycholic acid (CDCA), lithocholic acid (LCA), taurochenodeoxycholic acid (TCDCA), ursodeoxycholic acid (UDCA), and tauroursodeoxycholic acid (TUDCA) [13,14,15,16,17,18] (Figure 1). 

We comprehensively analyzed TGR5’s affinity energy with a number of bile acids using Autodock software (Figure 2). The results show that TGR5’s affinity energy with INT-777, CDCA, LCA, TCDCA, UDCA, and TUDCA is −8.8, −7.5, −6.7, −8.2, −7.5, and −7.7 kcal/mol, respectively. Our data demonstrate why INT-777 can be widely used to study TGR5. INT-777, as a special TGR5 agonist, reduced the severity of AP in mice, which was manifested as decreased pancreatic tissue damage as well as a decrease in the expression of serum enzymes (amylase and lipase), interleukin-1 beta (IL-1β), interleukin-6 (IL-6), tumor necrosis factor-α (TNF-α), and necrosis-related proteins (RIP3 and p-MLKL) caused by inhibiting the reactive oxygen species/nucleotide-binding oligomerization domain (NOD)-like receptor containing the pyrin domain (3ROS/NLRP3) inflammasome pathway [19]. In 3T3-L1 cells, CDCA remarkably inhibits ligand-stimulated peroxisome proliferator-activated receptor γ (PPARγ) transcriptional activity to decrease adipocyte differentiation [20]. LCA alleviates high-glucose-induced cardiac hypertrophy via enhancing sarcoplasmic/endoplasmic reticulum Ca^2+^ ATPase 2a (SERCA2a) and phosphorylated phospholamban (PLN) expression in H9C2 cells, thereby activating TGR5 [21]. The activation of TGR5 by TCDCA decreases TNF-α, IL-1β, IL-6, interleukin-8 (IL-8), and interleukin-12 (IL-12) expression through the cAMP-PKA-cAMP response element-binding protein (CREB) and Raf1-CREB signaling pathways in NR8383 cells [22]. UDCA can increase the transforming growth factor-β (TGF-β) ubiquitination level at the site of K135 by means of Hsc70-interacting protein (CHIP) and phosphorylate TGF-β at the T282 site via the TGR5-cAMP-PKA axis, commonly causing antitumor immunity [23]. TUDCA decreases DNA-dependent protein kinase (DNA-PK), tumor suppressor P53-binding protein-1 (53BP-1), and DNA ligase IV expression levels to impair DNA damage in embryos via activating the TGR5 receptor, thereby reducing endoplasmic reticulum (ER) stress [24]. However, a large array of evidence has shown that TGR5 is implicated in fatty metabolism, stress regulation, inflammation, and immunity. Hence, numerous results have revealed that the activation of TGR5 can prevent the development and migration of certain tumors; however, TGR5 is the opponent in other tumors. 

## 2. TGR5 in Lung Cancer

Lung cancer is the leading cause of cancer deaths worldwide. In China, lung cancer is also one of the most common malignant tumors, accounting for 25% of the death toll of malignant tumors [25]. Lung cancer includes non-small cell, small cell, mesothelioma, thymoma, and neuroendocrine tumors [26]. Tests to diagnose lung cancer mainly include imaging tests, sputum cytology, and taking tissue samples (biopsy). Nowadays, a series of drugs and treatment methods can be used for the therapy as well as diagnosis of lung cancer, such as radiotherapy, chemotherapy, immunochemotherapy, molecular targeted therapy, Chinese medicine therapy, sotorasib (AMG-510), rybrevant (amivantamab-vmjw), capmatinib (tabrecta), lorlatinib (lorbrena), libtayo (cemiplimab), tepotinib (tepmetko), cisplatin, cyclophosphamide, and 5-FU [27,28,29,30,31,32]. However, many drugs directly or indirectly lead to severe adverse reactions, such as nausea, vomiting, and constipation [33]. Currently, Chinese traditional medicine is more widely used to regulate and treat physiological and pathological processes, such as dihydroartemisinin (DHA), which regulates immune cell heterogeneity by triggering a cascade reaction of cyclin-dependent kinases (CDK) and mitogen-activated protein kinase (MAPK) phosphorylation [34]; salvia miltiorrhiza and pueraria lobata, two eminent herbs in Xin-Ke-Shu (XKS), which ameliorate myocardial ischemia partially by modulating the accumulation of free fatty acids in rats [35]; and a monomeric polysaccharide from polygonatum sibiricum, which improved cognitive function in a model of Alzheimer’s disease by reshaping the gut microbiota [36]. More and more research results have revealed that TCDCA, as one of the main active ingredients of bile acids, after combining and activating TGR5 in lung cancer cells of H1299, can increase cAMP content and elevate phosphorylation levels of protein kinase A (PKA) and CREB [22]. In addition, PKA can phosphorylate many downstream kinases, such as Raf, glycogen synthase kinase-3 (GSK3), and focal adhesion kinase (FAK), while the activation of the cAMP-PKA-CREB signaling pathway may promote lung cancer cell growth, migration, invasion, and metabolism [37]. Verma demonstrated that the expression of the dominant negative form of PKA (dnPKA) or treatment with the PKA-specific inhibitor, H89, greatly reduced the growth of small cell lung cancer (SCLC) tumors [38]. Hye-Sook Seo found that the expression levels of mRNA and CREB and phosphorylated CREB (pCREB) were significantly higher in most of the non-small cell lung cancer (NSCLC) cell lines than in the normal human tracheobronchial epithelial (NHTBE) cells and adjacent normal lung tissue, respectively [39]. It is well known that NF-κB is a downstream protein of CREB, and the suppression of NF-κB activity can remarkably augment the development of tumor specimens in SCLC tumors. Additionally, a previous study by the present authors demonstrated that TGR5 activation strongly inhibited the Janus kinase-2 (JAK2)-STAT3 signaling pathway in vitro and in vivo. Jiang reported that the activation of TGR5 on membranes in NSCLC cell lines mediates the JAK2-STAT3 signaling pathway, exacerbating tumor cell development and migration [40]. Altogether, our summarized results strongly suggest that the cAMP-PKA-CREB and JAK2-STAT3 signaling pathways induced by TGR5 are promising therapeutic strategies and predict the effects of therapy for lung cancer (Figure 3).

## 3. TGR5 in Liver Cancer

Liver cancer is generally classified as primary or secondary. Primary cancer is commonly manifested in a malignant tumor beginning in the cells of the liver [41]. Researchers have generally divided primary cancer into three types: hepatocellular carcinoma (HCC) (hepatoma), cholangiocarcinoma (CCA), and angiosarcoma [42,43,44]. The latest statistics show that more than 900,000 people are diagnosed with HCC every year around the world, and HCC is a leading cause of cancer death worldwide, accounting for more than 800,000 deaths each year—thus ranking as the third most common cause of cancer-related death. HCC is more common in people who drink large amounts of alcohol and who have an accumulation of fat in the liver. Nowadays, blood tests, image tests (CT and MRI), and liver biopsies are usually used to diagnose HCC [45]. These methods can produce a range of adverse reactions in liver cancer patients. HCC treatments include surgery, liver transplant surgery, destroying cancer cells with heat or cold, delivering chemotherapy or radiation directly to cancer cells, radiation therapy, targeted drug therapy, immunotherapy, and clinical trials [46]. However, a great number of treatment and diagnosis methods of HCC definitely bring about several adverse reactions, such as an abdominal mass or lump, right-sided abdominal pain, right-shoulder blade pain, jaundice, itching, bloating, shortness of breath, unintentional weight loss and gain, loss of appetite, nausea and vomiting, fatigue and weakness, fever, and a general feeling of being unwell [47]. With a large number of people worldwide being infected by COVID-19, many traditional Chinese medicines have been used against the coronavirus disease (COVID-19), such as Jinhua Qinggan granule, Lianhua Qingwen capsule, Xuebijing injection, a lung cleansing and detoxifying decoction, and Huashibaidu formula, etc. [47,48,49,50,51]. In recent years, bile acids, as one of the main active components of bile, have been used for experimental research into liver cancer via multiple pathways. Mounting research results have demonstrated that TGR5 is one of the most common therapy targets against HCC via the regulation of energy homeostasis and glucose metabolism. The α7-nicotinic acetylcholine receptor (α7-nAChR) is an oncogene and risk factor for HCC. In HCC, knocking-down α7-nAChR can reduce cell viability, inhibit cellular proliferation, attenuate migration and invasion, and diminish the sphere-formation ability of HCC, which is related to the phosphorylation of JAK2, STAT3, Ras homolog family member A (RhoA), Rho-associated protein kinase-1 (ROCK1), matrix metallopeptidase-2 (MMP2), and matrix metallopeptidase-9 (MMP9) in HCC—mediated by TGR5 [52]. However, the expression of matrix metalloproteinases (MMPs) is also higher in TGR5−/− mice, which may promote the development and migration of HCC [53]. In vivo, a lack of TGR5 in mice can promote diethylnitrosamine (DEN)-induced hepatocyte death, compensatory proliferation, and the gene expression of certain inflammatory cytokines, matrix metalloproteinasesacute, and liver carcinogenesis to a greater extent than wild-type (WT) mice; in vitro, TGR5 activation strongly inhibited the proliferation and migration of HCC via suppressing STAT3 signaling, and its DNA binding activity [54]. In summary, TGR5 receptor could be a new potential biomarker for the diagnosis and treatment of HCC in the future (Figure 4).

## 4. TGR5 in Gastric Cancer

Although the occurrence of gastric cancer, also commonly called stomach cancer, has declined significantly over the past two decades, it is still among the most prevalent cancers worldwide [55,56]. According to recent research, the occurrence of gastric cancer relates to diet [57]. Helicobacter pylori infecting people’s stomachs could become a special biomarker for the diagnosis and therapy of gastric cancer [58]. It can directly cause chronic gastric inflammation, which slowly progresses to atrophy, metaplasia, dysplasia, and gastric cancer when H. pylori enters into the stomach [59]. There are 1,000,000 people diagnosed with gastric cancer worldwide every year, who are mainly distributed in countries in East Asia, Eastern Europe, and Central and South America [45]. Gastric cancer patients normally experience bloating after eating, heartburn, a lack of appetite, nausea, and an upset stomach [60]. Tumor removal and chemotherapy are the foremost treatment methods for gastric cancer patients [61]. However, these methods pose a risk and garner great cost for gastric cancer patients and their families. TGR5 can be activated by traditional Chinese medicines, thereby mediating a great number of signal transduction pathways for the treatment of gastric cancer, such as 23(S)-mCDCA. TGR5 overexpression in the gastric cancer cell line SGC7901 activated by 23(S)-mCDCA greatly inhibited the gene expression of interferon-inducible protein 10 (IP10), TNF-α, and chemoattractant protein-1 (MCP1) induced by NF-κB, as well as LPS. Sirtuin-1 (SIRT1), a class-III protein deacetylase, regulates cell death and metabolism via multiple physiological effects, such as DNA damage, anti-inflammation, and cellular oxidative stress [62,63]. The activation of SIRT1 not only inhibited the mRNA expression of STAT3 and c-Myc, but also suppressed the phosphorylation of NF-κB p65 [64,65,66]. Thus, SIRT1 has a repressive function on gastric cancer via inhibiting the activation of STAT3 and NF-κB. However, TGR5 activation strongly antagonizes the STAT3 signal pathway through suppressing the phosphorylation of STAT3 and its transcription activity induced by LPS. The activation of TGR5 on the gastric cancer cell line SGC7901 by 23(S)-mCDCA significantly inhibited the downstream gene expression of STAT3, as well as MMP2, complement component 3 (C3), c-Myc, interleukin 6 receptor (IL-6R), epidermal growth factor receptor (EGFR), endothelial PAS domain protein 1 (EPAS), suppressor of cytokine signaling 3 (SOCS3), MMP7, and MMP14 [67]. In summary, TGR5 activation can inhibit the proliferation and migration of gastric cancer cells via the suppression of STAT3 and NF-κB signal pathways, and thus TGR5 could be used to diagnose and treat gastric cancer in the future (Figure 5). 

## 5. TGR5 in Colorectal Cancer

Colorectal cancer (CRC), sometimes called colon cancer, results in 900,000 deaths every year and has been considered the fourth leading cause of mortality related to cancer diseases worldwide [45]. In 2020 in the USA, approximately 147,950 individuals were diagnosed with CRC and 53,200 died from the disease, including 17,930 cases and 3640 deaths in individuals aged younger than 50 years [68]. CRC development in patients can lead to the occurrence of ulcerative colitis and Crohn’s disease with age [69]. Mounting evidence shows that the risk factors for CRC include a lack of regular physical activity, a diet low in fruit and vegetables, a low-fiber and high-fat diet, being overweight or obese, alcohol consumption, and tobacco use, etc. [70]. Today, surgery, radiation therapy, and chemotherapy are the key components of CRC therapy [71]. CRC therapy can cause a range of side effects, such as nausea, vomiting, loss of appetite, diarrhea, abdominal pain, etc. [68]. Bile acids are one of the main components of bile, exerting anti-inflammation and immune functions via the cAMP-PKA-CREB and Raf1-CREB signaling pathways mediated by TGR5 [22]. Continuous primary inflammation might trigger the production and development of CRC. In HCT116 cells, SW480 cells, and DSS-induced CRC mice, UDCA, as one of the main active components of bile, suppresses the malignant progression of CRC via TGR5 mediating the cAMP-PKA-RhoA signal pathway to antagonize Yes-associated protein (YAP) [72]. Moreover, LCA remarkably activates TGR5 to repress the production of pro-inflammation cytokines in the colon, decreasing the development and migration of CRC [73]. Taken together, TGR5 combined and activated by bile acids is considered as a novel treatment target for CRC (Figure 6). 

## 6. TGR5 in Other Cancers

The membrane bile acid receptor TGR5 is a multiple-target receptor, and it can mediate several cell signal pathways to regulate itching, inflammation, proliferation, migration, insulin release, monocyte adhesion, and muscle relaxation, such as NF-κB, ERK, AKT, STAT3, transient receptor potential ankyrin 1 (TRPA1), exchange protein activated by cAMP (Epac), and RhoA [4,74,75,76]. Therefore, TGR5 is not only a metabolism regulator, but also has multiple functions in other cancers, such as endometrial cancer, breast cancer, and pancreatic cancer. First, CDCA at low concentrations significantly inhibited Ishikawa cell growth by inducing a remarkable increase in cyclin D1 protein and mRNA expression via TGR5 mediating the ERK-CREB signal pathway, suggesting that CDCA activated the TGR5-dependent CREB signaling pathway to promote human endometrial cancer cell proliferation [77]. Second, in MCF-7 and MDA-MB-231 breast cancer cells, LCA exerted anti-proliferative and pro-apoptotic effects through TGR5 mediating the cAMP-PKA-CREP signal pathway, thereby negatively regulating the gene and protein expressions of P53 and Bcl-2 to suppress the migration and development of breast cancer—suggesting that TGR5 is a novel target for interventions in breast cancer [78]. In addition, LCA can increase oxidative stress in breast cancer via TGR5. In the breast cancer cell line MCF-7, LCA decreased nuclear factor-2 (NRF2) expression and increased Kelch-like ECH associating protein 1 (KEAP1) expression via the activation of TGR5 and constitutive androstane receptor (CAR); in breast cancer patients, the overexpression of inducible nitric oxide synthases (iNOS), neuronal nitric oxide synthases (nNOS), chimeric antigen receptor (CAR), KEAP1, NADPH oxidase 4 (NOX4), and TGR5 or the downregulation of nuclear factor erythroid 2-related factor 2 (NRF2) were correlated with better survival, except for triple negative cases; therefore, TGR5 activation by LCA significantly suppressed the proliferation of breast cancer cells via oxidative stress [79]. Moreover, in pancreatic cancer, TGR5 had a significantly higher expression in the cancerous tissues than the adjacent normal tissues (81.6% vs. 36.8%), and Cox proportional hazards regression analysis confirmed that TGR5 expression was an independent predictor of the overall survival of patients with pancreatic cancer (*p* = 0.019), suggesting that TGR5 might serve as an important therapeutic target for pancreatic cancer [80] (Figure 7).

## 7. Conclusions

Since its identification in 2002, TGR5 has been found to be ubiquitously expressed in humans and animals, and is known to activate various intracellular signaling pathways upon interaction with bile acids. It has been continuously reported that the activation of TGR5 by bile acids mediated the cAMP-PKA-CREB, JAK2-STAT3, cAMP-PKA-RhoA, and cAMP-PKA-CREP signaling pathways, affecting the proliferation and migration of lung cancer, liver cancer, gastric cancer, colorectal cancer, endometrial cancer, breast cancer, and pancreatic cancer through regulating some special gene and protein expression, such as Raf, GSK3, FAK, Yap, cMyc, IL-6R, EGFR, EPAS, SOCS3, MMP7, and MMP14. In this review, we found that TGR5-dependent signaling pathways can promote the development and migration of lung cancer, endometrial cancer, and pancreatic cancer; meanwhile, it can inhibit the proliferation and migration of liver cancer, gastric cancer, colorectal cancer, and breast cancer. Hence, TGR5 has double-regulatory functions in the development process of cancer.

## Figures and Tables

**Figure 1 molecules-27-05292-f001:**
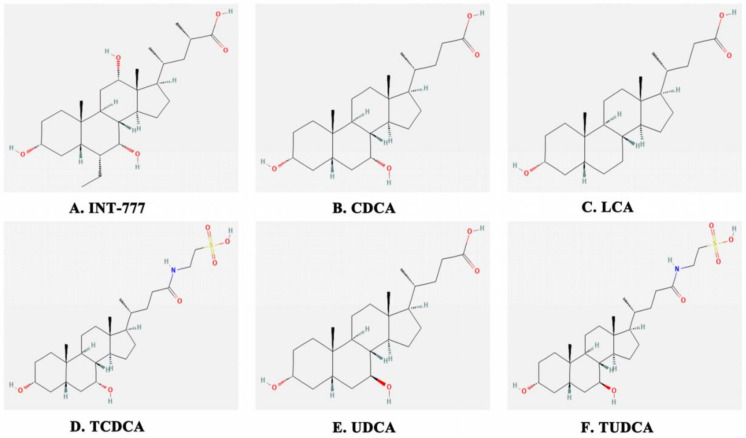
The chemical structures of the bile acids.

**Figure 2 molecules-27-05292-f002:**
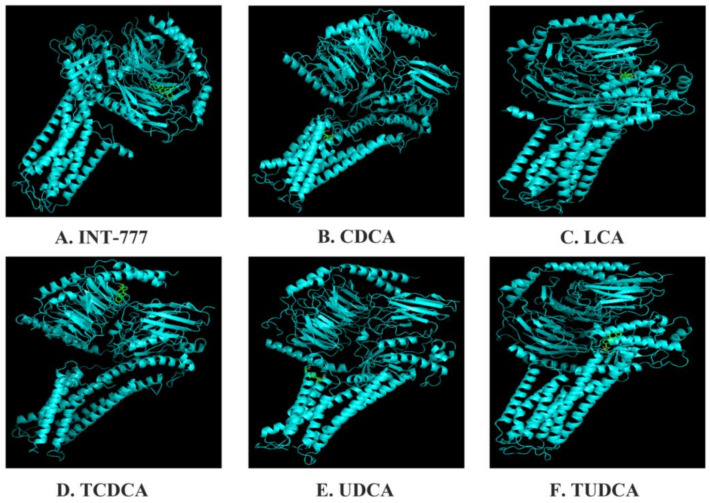
Molecular docking of TGR5 to bile acids.

**Figure 3 molecules-27-05292-f003:**
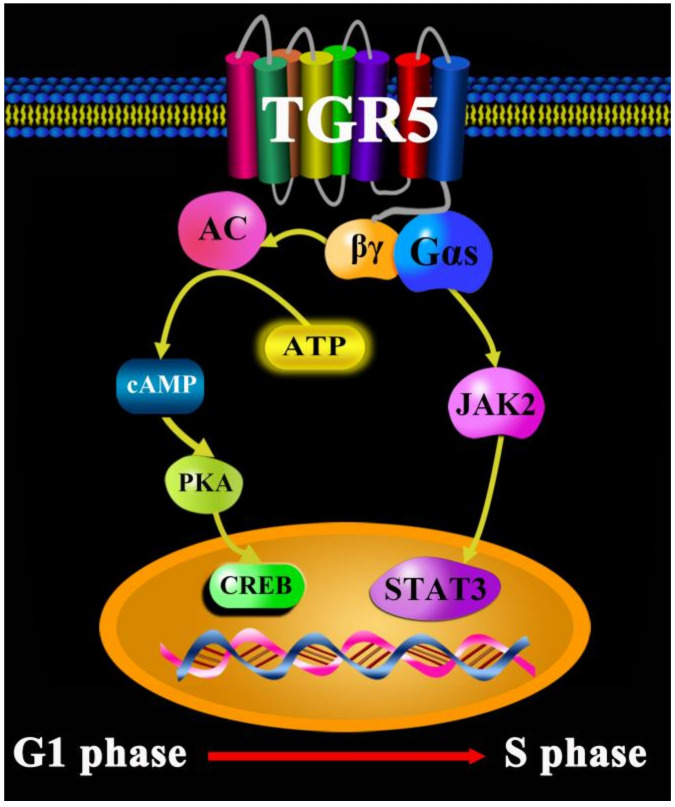
TGR5 in lung cancer.

**Figure 4 molecules-27-05292-f004:**
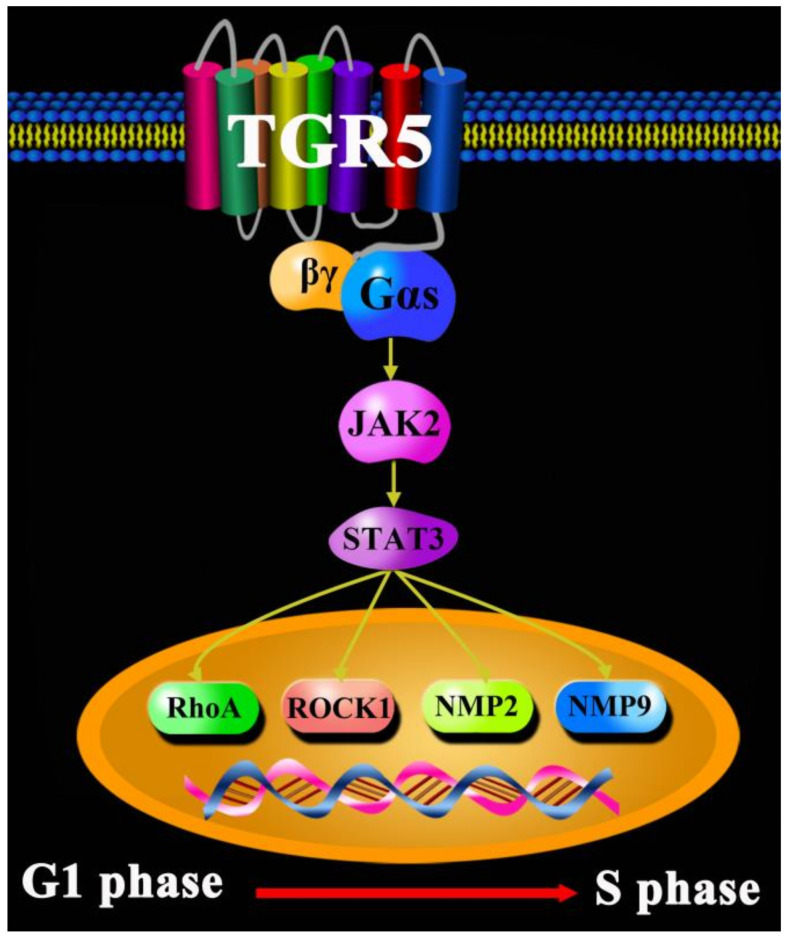
TGR5 in liver cancer.

**Figure 5 molecules-27-05292-f005:**
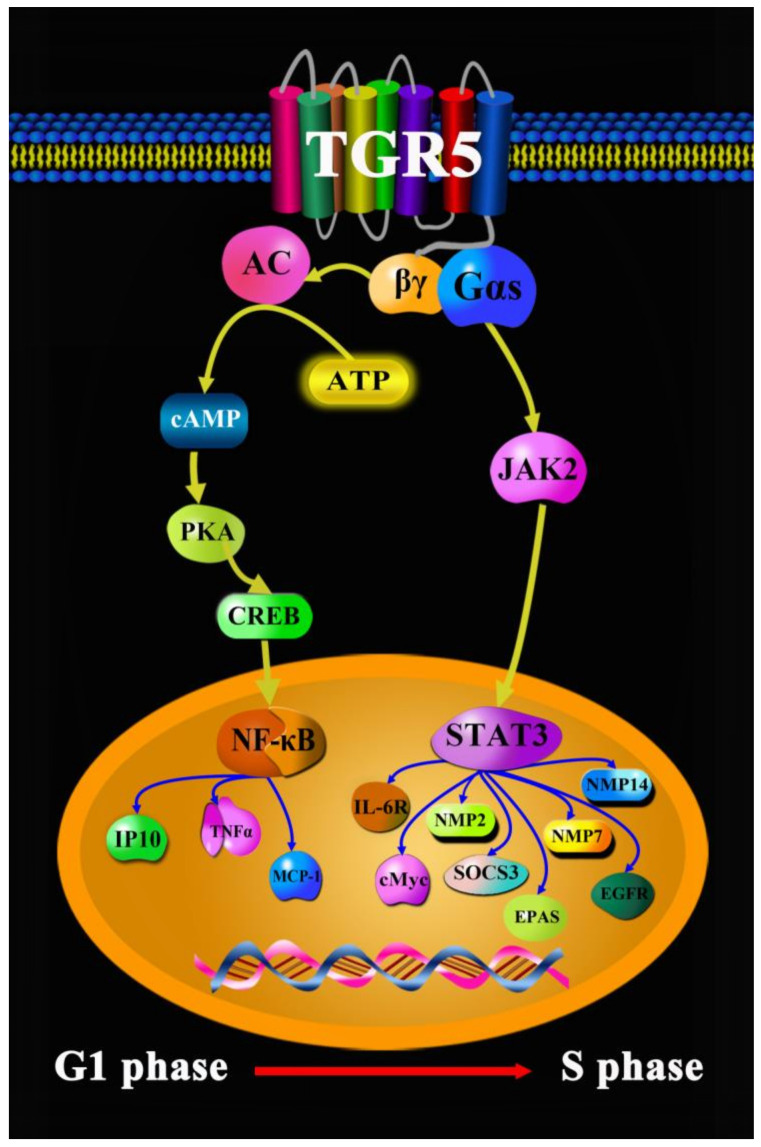
TGR5 in gastric cancer.

**Figure 6 molecules-27-05292-f006:**
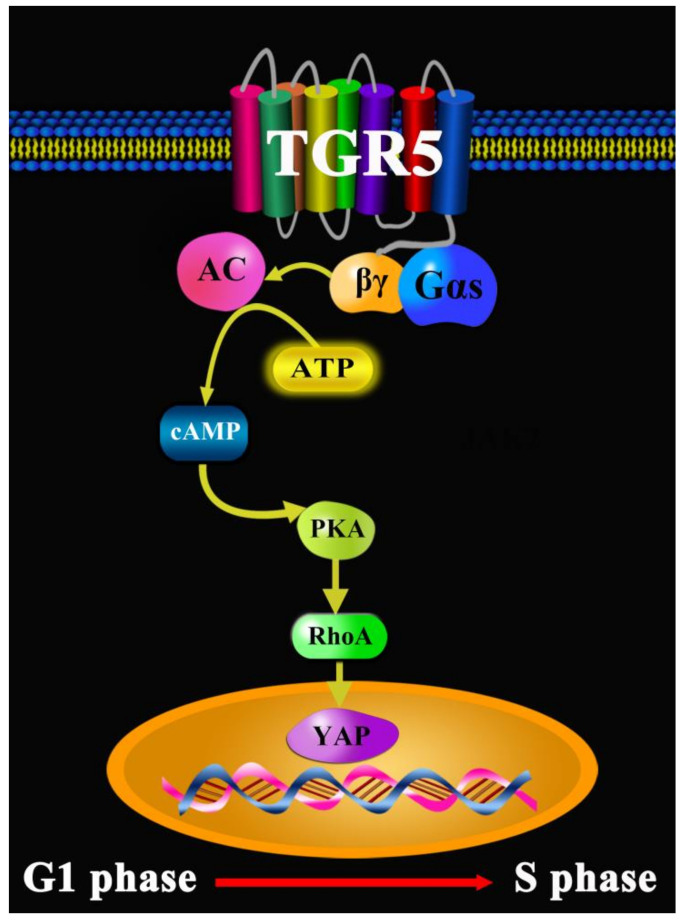
TGR5 in colorectal cancer.

**Figure 7 molecules-27-05292-f007:**
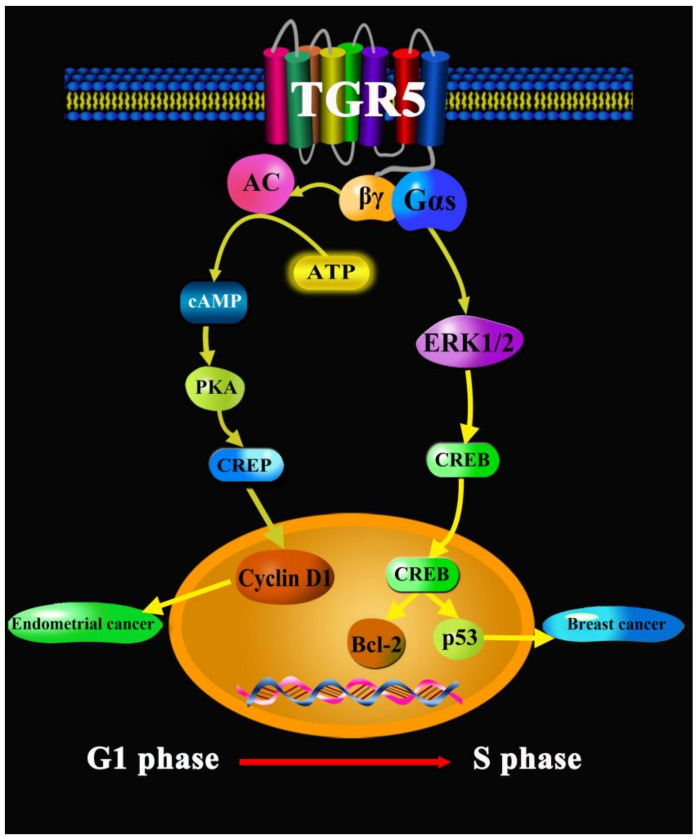
TGR5 in other cancers.

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
