# Peer review of "The Bile Acid Membrane Receptor TGR5 in Cancer: Friend or Foe?"

_molecules, 2022, doi:10.3390/molecules27165292_

Round 1

Reviewer 1 Report

The authors of the manuscript review interesting and lately quite important membrane receptor TGR5 and its relation to various types of cancer. The review touches the important areas and evaluates positive and negative effects related to TGR5. Nevertheless, to increase the clarity of the review I would suggest major language corrections and additional formatting changes, e.g., introducing paragraphs in the text to separate different thematic areas.

There are also many typos throughout the text, for example:

line 43 – is belong

51 – lithocholic not chalic

57-58 – the sentence is not clear

72 – But, Numerous

73 – TGR5 is friendly in certain tumors – what does it mean please?

79 – diagnosis of lung…

127, 131, etc. – please move all the web-links from text to references

128 – These methods produce a certain of pain on body and physiology?

177 – TGR5 can activated? A great many?

211 – Bile acids is – introduction of abbreviations of UDCA and LCA is repeating (line 215 and 218)

220 – Take together?

225 – can mediates?

257 – TGR5 has been found to be ubiquitously expression?

References: journal names of majority of references are given in their abbreviated form but also many are not, please unite that

Author Response

Thank you for your letter and for the reviewers’ comments concerning our manuscript entitled ‘The bile acid membrane receptor TGR5 in cancer: friend or foe? (Manuscript ID molecules-1856347). These comments are all valuable and very helpful for revising and improving our paper, as well as the important guiding significance to our researches. We have studied comments carefully and have made correction which we hope to meet with approval. Revised portion are highlighted in yellow in the revised manuscript. The main correction in the manuscript and the responds to the reviewers’ comments are following:

Responds to the reviewers’ comments:

Reviewer #1:

  1. Response to comment: line 43 – is belong 

Response:According to review’s suggestion, we carefully checked the word “is belong to” is correct.

  1. Response to comment: 51 – lithocholic not chalic

Response:According to review’s suggestion, we carefully checked the word “6α-ethyl-23(S)-methyl-cholic acid” is correct.

  1. Response to comment:57-58 – the sentence is not clear

Response:According to review’s suggestion, we wrote the long sentence again, “INT-777, as a special TGR5 agonist, could reduce the severity of AP in mice, which was manifested as decreased pancreatic tissue damage as well as the decrease of serum enzymes (amylase and lipase), interleukin-1 beta (IL-1β), interleukin-6 (IL-6), tumor necrosis factor-α (TNF-α), and necrosis related proteins (RIP3 and p-MLKL) expression by inhibiting reactive oxygen species/nucleotide-binding oligomerization domain (NOD)-like receptor containing pyrin domain (3ROS/NLRP3) inflammasome pathway.”

  1. Response to comment: 72 – But, Numerous

Response: We deleted the word “but” in manuscript.

  1. 5. Response to comment: 73 – TGR5 is friendly in certain tumors – what does it mean please?

Response: Activation of TGR5 can prevent the development and migration in  certain tumors.

  1. Response to comment: 79 – diagnosis of lung…

Response: Tests to diagnose lung cancer mainly include imaging tests, Sputum cytology, and tissue sample (biopsy).

  1. Response to comment:127, 131, etc. – please move all the web-links from text to references.

Response: We have moved all the web-links from text to references.

  1. Response to comment:128 – These methods produce a certain of pain on body and physiology?

Response: These methods definitely produce a certain of adverse reactions to liver cancer patients.

  1. Response to comment:177 – TGR5 can activated? A great many?

Response: TGR5 can be activated by traditional Chinese medicines mediating a great number of signal transduction pathways for treatment of gastric cancer.

  1. Response to comment:Bile acids is – introduction of abbreviations of UDCA and LCA is repeating (line 215 and 218).

Response: we have changed all the bile acids for its abbreviations according to referee’s suggestions.

  1. Response to comment:Take together?

Response: we have changed “take together” for “taken together”.

  1. Response to comment:225 – can mediates?

Responsewe have changed “can mediates” for “can mediate”.

  1. Response to comment:257 – TGR5 has been found to be ubiquitously expression?
  2. Response: Since its identification in 2002, TGR5 has been found to be ubiquitously expressed in humans and animals, and to activate various intracellular signalling pathways upon interaction with bile acids.
  3. Response to comment:References: journal names of majority of references are given in their abbreviated form but also many are not, please unite that.

Response: we have checked out all references and made them unite.

 Special thanks to you for your good comments.

We tried our best to improve the manuscript and made some changes in the manuscript. These changes will not influence the content and framework of the paper. And we have highlighted in yellow in revised paper.
We appreciate for Editor/Reviewer’s warm work earnestly, and hope that the correction will meet with approval.
Once again, thank you very much for your comments and suggestions.

Best regards,

Youchao Qi

Reviewer 2 Report

The G protein-coupled bile acid receptor 1 (GPBAR1) also known G-protein coupled receptor 19 (GPCR19), membrane-type receptor for bile acids (M-BAR) or TGR5 as is a protein that in humans is encoded by the GPBAR1 gene. The role of TGR5 in the regulation of metabolism is well established. The authors of this manuscript provide an overview of the role of TGR5 in neoplasmic transformation, which certainly puts TGR5 in the spotlight.

In the review, Authors found that TGR5-dependent signalling pathways can promote development and migration of

·         lung cancer,

·         endometrial cancer,

·         and pancreatic cancer.

On the other handit It can inhibit proliferation and migration of liver cancer, gastric cancer, colorectal cancer, and breast cancer. Thence, TGR5 has double-regulation functions in the development process of cancer .

Strength:

The significance of the manuscript is that it is richly illustrated with schemes of signal cascades, which makes it much easier to follow the matter.

Weakness:

The authors could provide the molecular structures of bile acids, state that INT-777 is a synthetic bile acid, perhaps discuss the binding energies (or efficiency) of various bile acids to TGR5, provide some docking images, which structural element of bile acids is most important for binding for TGR5.

Author Response

Thank you for your letter and for the reviewers’ comments concerning our manuscript entitled ‘The bile acid membrane receptor TGR5 in cancer: friend or foe? (Manuscript ID molecules-1856347). These comments are all valuable and very helpful for revising and improving our paper, as well as the important guiding significance to our researches. We have studied comments carefully and have made correction which we hope to meet with approval. Revised portion are highlighted in yellow in the revised manuscript. The main correction in the manuscript and the responds to the reviewers’ comments are following:

Responds to the reviewer’s comments:

 Reviewer #2:

  1. Response to comment: The authors could provide the molecular structures of bile acids, state that INT-777 is a synthetic bile acid, perhaps discuss the binding energies (or efficiency) of various bile acids to TGR5, provide some docking images, which structural element of bile acids is most important for binding for TGR5.

Response: We totally thought that it is a great suggestion to enrich our manuscript.

Special thanks to you for your good comments.

We tried our best to improve the manuscript and made some changes in the manuscript. These changes will not influence the content and framework of the paper. And we have highlighted in yellow in revised paper.

We appreciate for Editor/Reviewer’s warm work earnestly, and hope that the correction will meet with approval.

Once again, thank you very much for your comments and suggestions.

Best regards,

Youchao Qi

Round 2

Reviewer 1 Report

Dear authors, thank you for the careful corrections. Out of the comments corrected, I still would like to come back to these two:

  1. Response to comment: line 43 – is belong 

ResponseAccording to review’s suggestion, we carefully checked the word “is belong to” is correct.

"belongs to" is correct

  1. Response to comment: 51 – lithocholic not chalic

ResponseAccording to review’s suggestion, we carefully checked the word “6α-ethyl-23(S)-methyl-cholic acid” is correct.

Line 51 - I meant to correct "lithocholic acid" instead of surrent typo "lithochalic acid"

Author Response

Dear  reviewer:

Thank you for your comments again on our manuscript entitled ‘the bile acid membrane receptor TGR5 in cancer: friend or foe?’ (Manuscript ID molecules-1856347). First, we exceedingly thank the reviewer’s  comments, and revised the mistake in the revised manuscript. Second, we carefully checked the manuscript and highlighted all mistakes in yellow.

Last, we hope our manuscript could successfully publish in the journal of molecules.

On behalf of all authors of the manuscript, I would to thank all the suggestions and help of reviewer’s to our manuscript, wish you work well and happy every day.

Best regards,

Youchao Qi